# Targeting Microglia-Synapse Interactions in Alzheimer’s Disease

**DOI:** 10.3390/ijms22052342

**Published:** 2021-02-26

**Authors:** Gaia Piccioni, Dalila Mango, Amira Saidi, Massimo Corbo, Robert Nisticò

**Affiliations:** 1Laboratory Pharmacology of Synaptic Plasticity, European Brain Research Institute, 00161 Rome, Italy; d.mango@ebri.it (D.M.); mouraissame@hotmail.com (A.S.); 2Department of Physiology and Pharmacology “V.Erspamer”, Sapienza University of Rome, 00185 Rome, Italy; 3School of Pharmacy, University of Rome “Tor Vergata”, 00133 Rome, Italy; 4Department of Neurorehabilitation Sciences, Casa Cura Policlinico, 20144 Milan, Italy; m.corbo@ccppdezza.it

**Keywords:** microglia, Alzheimer’s disease, synaptic plasticity, neuroinflammation, long-term potentiation, cytokines

## Abstract

In this review, we focus on the emerging roles of microglia in the brain, with particular attention to synaptic plasticity in health and disease. We present evidence that ramified microglia, classically believed to be “resting” (i.e., inactive), are instead strongly implicated in dynamic and plastic processes. Indeed, there is an intimate relationship between microglia and neurons at synapses which modulates activity-dependent functional and structural plasticity through the release of cytokines and growth factors. These roles are indispensable to brain development and cognitive function. Therefore, approaches aimed at maintaining the ramified state of microglia might be critical to ensure normal synaptic plasticity and cognition. On the other hand, inflammatory signals associated with Alzheimer’s disease are able to modify the ramified morphology of microglia, thus leading to synapse loss and dysfunction, as well as cognitive impairment. In this context, we highlight microglial TREM2 and CSF1R as emerging targets for disease-modifying therapy in Alzheimer’s disease (AD) and other neurodegenerative disorders.

## 1. Introduction

Synaptic plasticity refers to the capability of experience to modify neural circuit function and thereby influence thinking, feeling, and behavioral patterns. Long-term potentiation (LTP) and long-term depression (LTD) of synaptic transmission represent the principal experimental model for the synaptic changes underlying learning and memory [1,2]. It is widely recognized that alterations in normal synaptic function are not only a core feature, but also a leading cause of several neuropsychiatric diseases, including Alzheimer’s disease (AD) [3].

Microglia, a specialized population of cells present in the central nervous system, are considered immune sentinels which mediate a potent inflammatory response, but are also involved in many central processes as synaptic organization, trophic neuronal support during development, and the control of neuronal excitability [4].

Neuroinflammatory stimuli that occur in neurodegenerative process can alter synaptic plasticity through alteration of microglia immune-related pathways [5]. Indeed, the close interactions between microglia and synapses lead to the so-called synaptic stripping hypothesis [6], a process in which microglia can selectively eliminate dysfunctional synapses. This microglia-mediated synapse removal, normally associated with activity-dependent refinement during neurodevelopment, can be reactivated in aging or in neurodegenerative diseases [7].

The present review focuses on the emerging cellular and molecular mechanisms linking changes in microglia functionality to synaptic alterations in AD, and highlights the microglia–synapse interaction as a potential target for the treatment and prevention of AD.

## 2. The Synaptic Basis of AD

Synaptic plasticity has certainly contributed to our understanding of various diseases that affect cognition. This has been typified by work on AD. Identification of genetic mutations associated with familial AD led over the past years to the generation of numerous transgenic animal models with different characteristics. Most of these express human amyloid precursor proteins (APP) and presenilin (PS1, PS2) mutations. To circumvent the drawbacks of the first-generation models, single humanized App knock-in (KI) mice were then generated [8,9].

Currently studied models show cognitive deficits, age-related disruption of synaptic markers, and amyloid plaque deposition, but few strains show evidence of significant cell death [10,11]. Since loss of memory is one of the major hallmarks of the disorder, the phenotypic characterization of these animals has classically included electrophysiological studies to analyze synaptic transmission and LTP/LTD in the hippocampus.

Most of studies have reported, principally, either inhibition of LTP or reduction in baseline fast excitatory transmission [12,13] prior to plaque deposition, as well as amplification of LTD [14,15,16,17].

Unfortunately, very few findings obtained in animal models have resulted in target validation or led to successful translation into disease-modifying compounds in AD patients [18,19].

In addition to amyloid beta (Aβ) plaques and neurofibrillary tangles (NFT), the brain of patients with AD manifests a sustained inflammatory response [20]. Neuroinflammation has been observed not only in post-mortem AD tissues [21], but also in the different animal models of AD [22,23,24,25].

It was originally thought that a persistent inflammatory response in the brain of AD patients was the consequence of the neuronal loss associated with this disorder. More recent studies have suggested that a sustained immune response in the brain facilitates and aggravates both Aβ and NFT pathologies and neurodegeneration [26,27]. It has also been proposed that neuroinflammation provides a link between the early Aβ pathology and subsequent NFT formation [28].

Among the different proinflammatory mediators involved in AD, tumor necrosis factor (TNF) α plays a central role at the synaptic level [29]. Indeed, this cytokine mediates the disrupting effects of Aβ on LTP in animal models of AD. On the other hand, normal LTP following Aβ exposure was observed either in transgenic mice lacking TNF receptor type 1 or in the presence of anti-TNF agents such as infliximab and thalidomide [30].

Likewise, the proinflammatory cytokine interleukin (IL) 1β is also involved in the synaptotoxic effect mediated by Aβ oligomers. Indeed, the interleukin-1 receptor antagonist (IL-1Ra) rescues LTP impairment alteration following Aβ application [31].

However, the role of cytokines in AD is rather complex considering their anti-inflammatory or pro-inflammatory profiles.

Several approaches targeting neuroinflammation in AD models have been undertaken. Among these, nonsteroidal anti-inflammatory drugs (NSAIDs) can exert neuroprotection through inhibition of inflammatory events and suppression of early accumulation of amyloid pathology [32]. Of note, cyclooxygenase-2 (COX-2) inhibitors reversed LTP loss following soluble Aβ oligomers [33]. Moreover, ibuprofen attenuated early memory decline in AD model, and this effect was associated with modulation of hippocampal gene expression in pathways involved in synaptic plasticity [34]. Overall, these studies indicate that NSAIDs exert neuroprotection and prevent memory loss, even though the mechanisms remain unclear [35].

Despite these preclinical evidences, clinical trials investigating different compounds with anti-inflammatory properties did not provide so far encouraging results [36].

## 3. Microglia and Neuroinflammation in AD

### 3.1. Microglia in Brain Physiology

Different populations of macrophages deal with heterogeneous functions in the maintenance of the brain homeostasis. Among them, microglia cells are the principal type located in the central nervous system (CNS) parenchyma, where they are connected with neurons, astrocytes and oligodendrocytes [37].

During brain development, microglial cells originate from blood-derived precursors and require colony-stimulating factor 1 receptor (CSF1R) signaling for their proliferation and survival [38]. These progenitors invade the neural tissue and are distributed throughout the CNS acquiring a ramified phenotype known as resting microglia [39].

Microglia are involved in several processes in both healthy and pathological brain. Specifically, it plays a crucial role in the maintenance of appropriate synaptic connections and neuronal plasticity. Indeed, through the development of the visual system, microglia prune out the presynaptic inputs that originate from the retinal ganglion cells (RGCs) into the dorsal lateral geniculate nucleus (LGN), such that each LGN neuron receive inputs from one RGCs [40]. This mechanism of elimination/pruning of unused and immature connections is thought to be responsible for the correct efficiency of neuronal transmission during brain development [38]. Synaptic turnover mediated by microglia has been also observed during adulthood, a mechanism by which microglia seems to be involved in maintenance of the physiological neuronal activity and synaptic plasticity by influencing the LTP and LTD process [41,42], even though the underlying mechanisms remain elusive.

During the developmental synaptic pruning process, microglia directly contact synapses via specific molecular pathways. The main one involves the classical complement cascade: C1q and C3 complement proteins localize to the afferent terminals that need to be removed, representing an “eat me” signal for microglia, which express the C3 receptor (C3R) [43].

In addition to synaptic refinement, microglial cells are specialized in maintaining cerebral homeostasis through the induction of immune responses. Under normal conditions, immune responses evoked by microglia and macrophages act in a coordinated manner to elicit the first line of defense against toxic insults from both internal and external sources [44]. Indeed, in response to pathologic stimuli, loss of homeostasis or tissue changes, microglia change their morphology, antigen presentation, and phagocytic and secretory activity [45]. In such conditions, microglia activation is driven by pro- or anti-inflammatory molecules that behave as damage- and pathogen-associated molecular patterns (DAMPs-PAMPs) [46]. These molecules bind to pattern recognition receptors (PRRs) expressed by microglia, thus signaling the presence of a CNS insult and initiating an immune response [47].

In the brain, microglia cells can exist in two different states, “resting” and “activated” microglia. The first is characterized by branched morphology and is present in healthy brains while the latter has an amoeboid morphology and is typical of unhealthy brains [48]. Resting microglia have a low expression of surface receptors, such as the complement receptor CD45, CD14, and Mac-1 (CD11b/CD18) [49]. Microglia cells sense microenvironmental changes and respond to pathogens and injuries by becoming ‘‘activated’’, a process through which they rapidly change their ramified morphology to an amoeboid phenotype and migrate to the lesioned site, where they phagocytize pathogens [50]. Traditionally, two distinct and opposite phenotypes, neurotoxic (M1) and neuroprotective (M2), are identifiable for “activated microglia” which differ in terms of receptor expression, effector function as well as cytokine and chemokine production [51]. Depending on the received stimuli, microglia can be classically or alternatively activated, thereby having opposite roles in the CNS. Several experiments investigated the different polarization of microglia, showing how the stimulation with lipopolysaccharide (LPS) or interferon (IFN) γ induces the activation of the neurotoxic M1 phenotype, whereas IL-4 or IL-13 induces the neuroprotective M2 activation [52].

The M1 polarization, often called “classical activation”, is a pro-inflammatory state induced mainly in response to injuries and infections and acts as the first line of tissue defense [53]. This activation causes inflammation and consequent cytotoxicity by release of reactive oxygen species (ROS), nitrogen reactive species (NRS), nitric oxide (NO), pro-inflammatory cytokines, and chemokines including TNF-α, IL-1β, IL-6, IL-12, and IL-18. Additionally, it is accompanied by impaired phagocytic capacity [54] and decrease of neurotrophic factors release.

In contrast, an anti-inflammatory phase is promptly initiated to antagonize the pro-inflammatory responses and restore tissue homeostasis [53]. Indeed, the M2 polarization, referred as “alternative activation”, is characterized by secretion of cytokines with anti-inflammatory activity such as IL-4, IL-10, IL-13, transforming growth factor (TGF) β, growth factors (insulin-like growth factor, IGF-1; fibroblast growth factor, FGF; Colony-stimulating factor, CSF1), and neurotrophic growth factor (brain-derived neurotrophic factor, BDNF; glial cell-derived neurotrophic factor, GDNF). Of interest, IL-4 and IL-13 present anti-inflammatory proprieties which could suppress the production of some of the pro-inflammatory cytokines produced by M1 phenotype [55] and reduce NO release, protecting against neuron injury induced by LPS [56]. In turn, IL-4 and IL-13 stimulate microglia- M2 phenotype [57] and cause the expression of arginase (Arg) 1, Ym-1, CD200R, IL-10, TGFβ, and Fizzl-1, which serve as specific markers for M2 microglia. In this way, microglia can be neuroprotective and neurosupportive via different mechanisms that include glutamate uptake, removal of dead cells and accumulation of abnormal proteins or production of neurotrophic factors [48] (Figure 1).

### 3.2. Microglia in Pathological Conditions

Increasing findings suggest the chronic activation of microglia is a common pathological feature of neurodegenerative disorders characterized by neuroinflammation, such as Alzheimer’s disease (AD), Parkinson’s disease (PD), and multiple sclerosis (MS).

A central role of microglia in the progression of AD was emphasized by the evidence that misfolded Aβ plaques act as DAMP and thus activate PRRs [58]. Soluble Aβ oligomers and Aβ fibrils bind to several receptors of microglia comprehending CD14, CD36, CD47, α6β1 integrin, class A scavenger receptor, and toll-like receptors (TLRs) [59]. This binding leads to the switch from the quiescent to the active state of microglia which cluster around extracellular plaques, limiting the growth and accumulation of plaques [60].

Therefore, during early stage of AD, clustered microglia has protective effects since it eliminates Aβ plaques, dying or dead cells by its phagocytic activity or by releasing proteases (insulin degrading enzyme, neprilysin, matrix metalloproteinase 9 and plasminogen) [61]. Activated microglia participate in the phagocytosis of Aβ preventing the deposition of Aβ and the formation of amyloid plaques. Microglia clustering plaques for phagocytosis of Aβ has characterized by M2 activation phenotype [62]. Moreover, these macrophages create a physical barrier that prevents plaques spreading [63].

Although early microglia-induced neuroinflammation is a protective response to toxic Aβ, chronic activation may be harmful. An increase in number and size of Aβ plaques results in the dysfunction of microglia in the brain, which is characterized by the overproduction of proinflammatory cytokines leading to synaptic damage. Therefore, the phagocytic activity of microglia is reduced by proinflammatory cytokines, like as IFN-γ, IL-1β, and TNF-α that shift microglia into the pro-inflammatory M1 phenotype [64] contributing to neurotoxicity and synapse loss. As described previously, microglia play a role in complement-mediated synaptic pruning; therefore, a reactivation of this mechanism could drive the progression of neurodegenerative diseases associated with synaptic loss [5] (Figure 1).

In this context, a promising therapeutic strategy could be to modulate microglial activity, promoting neuroprotective phenotype, and attenuating neurotoxic inflammatory stimuli [65]. Minocycline has been widely studied in recent years for its novel mechanism of action. Even if it is a semisynthetic long-acting second-generation tetracycline that is classically active against gram-negative and gram-positive bacteria through inhibition protein synthesis, minocycline is emerging as a potent anti-inflammatory, antiapoptotic, and neuroprotective drug in models of neurodegenerative diseases [66]. Minocycline has a dual mechanism by which could reduce cerebral inflammation and subsequent neuronal loss. It reduces the activation of pro-inflammatory microglial phenotype (M1) and decreases microglial production of pro-inflammatory cytokines (IL-1β, IL-6, TNF-α) and neurotrophic factors (nerve growth factor, NGF) induced by Aβ [67]. In conclusion, minocycline is able to reduce inflammation in neurodegenerative diseases modulating the pathological shift of microglia and the consequent production of pro-inflammatory responses.

Additionally, microglia cells seem to guide the pathogenesis of AD by active interaction with neurons, astrocytes and oligodendrocytes. Through secretion of IL-1α, TNF-α and C1q, activated microglia leads to the genesis of reactive astrocytes. These distorted cells, called A1 astrocytes, lose the ability to promote neuronal survival, outgrowth, synaptogenesis, and phagocytosis, and induce death of neurons and oligodendrocytes during disease. Using knockout mice lacking microglia, astrocytes failed to activate A1s, showing as reactive microglia are required to induce A1 reactive astrocytes in vivo [68].

In conclusion, even if microglia cells are necessary to immune response in the CNS, protracted microglia polarization is involved in the progression of neurodegenerative diseases.

From this point of view, inflammation is a causal component rather than a simple consequence of the neurodegeneration. Therefore, a deeper understanding of how microglia-mediated inflammation boosts AD neurodegeneration is crucial for the development of future therapeutic strategies.

#### Cytokines Involved in Pathological Neuroinflammation

Microglia cells, together with astrocytes, are the major source of cytokines in AD and, in turn, the cytokines can modulate microglia activation. Neuroinflammation responses and consequent release of cytokines in AD is driven by pathological accumulation of Aβ [54].

Aβ oligomers interact with proinflammatory cytokines to induce neuronal damage via pathways that involve the release of reactive oxygen species (ROS) [69] and NO through activation of nicotinamide adenine dinucleotide phosphate (NADPH) oxidase, myeloperoxidase, and inducible nitric oxide synthase (iNOS). Consequently, inflammatory mediators released by activated microglia, like as IL-1β, IL-6, TNF-α, chemokines, matrix metalloproteinase 2 (MMP2), NO, and nuclear factor kappa-B (NF-κB) contribute to neurodegeneration and myelin damage in neurodegenerative diseases [48]. The inflammatory response has been observed in preclinical model systems of AD. Even if current knowledge of the neuroinflammatory response in AD is based mainly on in vitro and animal studies, these results are confirmed by analysis of post-mortem cases of AD that evidenced an increase in the total number of microglial cells in cortical grey matter and white matter [70].

The cytokines released by activated microglia can exhibit pleiotropic functions and their involvement in the disease progression is not always immediate. Different effects can be observed based on cytokines concentration levels.

Early release of pro-inflammatory cytokines such as IL-1β and IL-6 contributes to maintain LTP, neural plasticity, brain homeostasis and plaques clearance [71]. On the contrary, sustained cytokines release due to chronic neuroinflammation can compromise brain tissue through inflammatory and atrophic effects on brain volume leading to neurodegeneration and cognitive deficits, typical of AD [72].

Several researches have demonstrated that in vitro IL-1β is released by activated microglia after stimulation with Aβ [73].

IL-1β has been identified in clustered microglia cells around amyloid plaques and seems to increase Aβ deposition by acting on APP expression and proteolysis [74]. Moreover, high levels of caspase-1 in AD patients’ brain increase IL-1β concentrations through the maturation of its pro-form [75].

Among cytokines, TNF-α and IL-1β have been shown to mediate the detrimental effects of Aβ oligomers on LTP. Indeed, suppression of LTP by Aβ was absent in mutant mice null for TNF receptor type 1 and was prevented by the monoclonal antibody infliximab, the TNF peptide antagonist, and thalidomide, the inhibitor of TNFα production [42].

It has been demonstrated that LTP is suppressed by IL-1β and that there is an inverse association between IL-1β concentration and LTP amplitude [76]. Based on evidence that this proinflammatory cytokine exerts an inhibitory effect on LTP in Cornu ammonis (CA) 1 [77], CA3 [78], and dentate gyrus [79], authors have analyzed the downstream effects of IL-1β levels in hippocampal tissue by using electrophysiological techniques. Immunoblot analysis revealed a correlation between release of IL-1β and increase activity of Jun N-terminal kinase (JNK) and p38. IL-1β activates these kinases that, in turn, induce cell damage cell death. IL-1β-induced activation of JNK and p38 leads to a decrease in glutamate release and might be responsible for the attenuation in both the early and later components of LTP.

Moreover, some of the IL-1β effects might arise from its stimulatory effect on reactive oxygen species (ROS) production in hippocampal tissue, underlined by the evidence that IL-1β-induced changes were blocked in rats treated with dietary manipulation with antioxidants vitamins E and C. Increase in ROS production in hippocampus of rats previously treated with IL-1β is a consequence of increased superoxide dismutase activity [76].

These results indicate a sequential effect. IL-1β stimulates activity of superoxide dismutase that increases ROS. The boosting of ROS causes activation of stress-activated kinases JNK and p38 which, in turn, may inhibit glutamate release and result in inhibition LTP.

Thus, proinflammatory cytokines, mainly IL-1β and TNF-α, can impair neuronal function before structural synaptic changes occur [54].

In conclusion, Aβ triggers microglia activation which then contributes to the release of cytokines which impair LTP. Inhibiting microglia activation can prevent the block of LTP induction by Aβ [42].

### 3.3. Modulation of Microglia Activation

The centrality of microglia in neurodegeneration and neuroinflammation led to the study of the effects of its modulation in healthy and pathological conditions. Indeed, the risk of developing late-onset AD seems to be influenced by different genetic factors and several immunoreceptors expressed in microglia, which control its activation and modulate the progression of neurodegenerative disease. Therefore, pinpointing specific activating and inhibitory receptors as potential targets for therapeutic intervention could be critical.

In this context, we will give an overview of genes expressed in microglia, and focus on the colony-stimulating factor 1 receptor (CSF1R) and triggering receptor expressed on myeloid cells 2 (TREM2) (see also Figure 1).

#### 3.3.1. Gene Expression Profiling

Genes potentially involved in microglia activation are not yet completely understood [80]. Genome-wide association studies (GWAS) have identified possible genetic risk factors for AD preferentially expressed in microglia, such as *CD33,* Inositol Polyphosphate-5-Phosphatase D *(INPP5D), CR1,* Granulin Precursor *(GRN)* and ATP Binding Cassette Subfamily A Member 7 (*ABCA7)* [81].

*CD33*, also known as sialic acid-binding immunoglobulin-like lectin (SIGLEC)-3, is a member of the SIGLEC family of lectins. Polymorphisms in the CD33 locus (rs3865444, rs3826656, and rs114282264) are associated with late-onset AD risk [82]. CD33 expression from microglia cells has been correlated with Aβ accumulation and is upregulated in patients [83].

*INPP5D* encodes the lipid phosphatase SH-2 containing inositol 5′ polyphosphatase 1 (SHIP-1) that inhibits phagocytosis in macrophages [84]. Therefore, *INPP5D* expression could be associated with higher SHIP-1 level and consequent reduction of phagocytosis, explaining the elevated AD risk [85].

*CR1* encodes a receptor for the complement factors C1q, C3b, and C4b widely expressed on microglia [86] that could be associated with aberrant activation of synaptic pruning [87].

A similar mechanism based on reactivation of complement-mediated pruning involves *GRN* deficiency. The rs5848T variant reduces the expression level of GRN [88] and increases the production of C1q and C3 with consequent enhancement of phagocytic elimination of synapses [89].

Several loss-of-function variants of ABCA7, such as rs3764650G, are associated with an increased risk of AD [90]. ABCA7 encodes a 12-pass transmembrane protein belonging to the adenosine triphosphate (ATP)-binding cassette transporter family involved in lipid transportation and phagocytosis regulation. A reduction of phagocytic activity against Aβ was observed in microglia in knockout mice [91].

Among other genetic factors, the role of the squamous cell carcinoma antigen (SCCA) protein has been highlighted as a possible inducer of microglia activation. Its expression is associated also with increase in SCCA protein expression that has been suggested to be involved in conversion of resting microglia to an activated form [92].

Although current knowledge about risk genes for pathological microglial activation in AD is not yet comprehensive, current findings are pointing to genetic modulations as a novel therapeutic approach.

#### 3.3.2. CSF1R

Colony-stimulating Factor 1 receptor (CSF1R) is a single pass type I membrane protein expressed in macrophages, microglia, and osteoclasts and has two natural ligands, colony-stimulating factor 1 (CSF1) and IL-34 [93]. Its activation controls the development, differentiation and survival of myeloid lineage cells in CNS. Experimental studies have shown that mice lacking CSF1R or its ligands present reduced densities of macrophages [94].

Whether the chronic microglial activation in neurodegenerative diseases is dependent on CSF1R signaling remains an open question.

Under normal conditions, microglia cells are the only macrophages expressing CSF1R in the brain [95]. For this reason, orally bioavailable inhibitors of CSF1R, such as PLX3397 (Pexidartinib) and PLX5622, which are able to cross the blood–brain barrier, exert a selective effect on microglial cells and have been extensively experimented.

In a study [93], administration of CSF1R inhibitors led to the elimination of ~99% of all microglia brain-wide in healthy adult mice previously treated with LPS, with no significant effect on cognition and behavior. Successive removal of this inhibitor repopulated the brain with new cells, then differentiating into microglia. Given the rapid microglia depletion after treatment, the authors proposed that blocking of CSF1R signaling leads to microglial cells death rather than inhibition of their proliferation.

Similarly, in APP/PS1 mice, prolonged inhibition of CSF1R showed a decrease in microglial proliferation and also prompted the shifting of microglial inflammatory profile to an anti-inflammatory phenotype [70]. Interestingly, chronic microglial elimination did not cause any modification of Aβ levels or plaques load but prevented the neuronal loss and reduced the overall inflammation [96].

These results were also confirmed in 5xfAD mice, where treatment with CSF1R inhibitors resulted in massive microglia elimination with no alteration of overall amyloid plaques burden [96]. To further investigate the microglia–Aβ relationship, [96] repeated the treatment with PLX3397 in 1.5-month-old 5xfAD mice, before amyloid-β plaques development. Results have shown that microglia did not exert any protective effect against amyloid-β accumulation at pre-plaque stage. However, those treatments correlated with improvement in contextual memory deficits. A correlation between inhibitor concentration and effect on microglia was shown: high doses of CSF1R inhibitors (1200 mg/kg chow) are required for microglial elimination while lower doses (300 mg/kg chow) only have an effect on CSF1R signaling. In a successive work, [97] demonstrated that Aβ plaques were formed exclusively in tissues where microglia had not been depleted by CSF1R inhibition, evidencing the specificity of this effect.

Through its modulation, the presented results further suggest a primary role of microglia in AD-related neurodegeneration and neuroinflammation. Microglia inhibition seems indeed to be a protective factor in murine disease models without however contributing to the clearance of amyloid-β or plaque deposition.

#### 3.3.3. TREM2

Triggering receptor expressed on myeloid cells 2 (TREM2) is a surface receptor belonging to the immunoglobulin family that is expressed by a subset of myeloid cells including monocytes, dendritic cells, osteoclasts, and tissue macrophages in peripheral tissue [98]. In the brain, TREM2 was found to be expressed only by the microglia [99].

Ligands of this receptor are anionic and zwitterion phospholipids and lipoproteins that carry out their signal through immunoreceptor DNAX activation protein (DAP12). The activation of TREM2 leads to DAP12 phosphorylation on cytoplasmatic tyrosine residue and the consequent triggering of downstream signaling mediators by spleen tyrosine kinase (Syk) leading to an increase in the intracellular Ca^2+^ [100].

The interest on TREM2 arises from the evidences that this receptor seems to regulate innate immune response [101] and have an essential role in the modulation of microglia functions [102]. TREM2 appears to be necessary for microglial survival, promoting phagocytosis of apoptotic neurons and retarding inflammation responses [101].

TREM2 inhibits neurotransmitters by blocking M2 microglia and this may reveal the potential mechanism by which TREM2 inhibits microglial inflammatory responses [103]. Indeed, researchers [104] have demonstrated in vitro that TREM2 inhibits microglia-mediated production of proinflammatory cytokines induced by LPS. In a subsequent work, the same group of researchers conducted a study in humans, in which a role of TREM2 in the maintenance of the brain homeostasis via promotion of tissue debris clearance was found [105]. This explains why the modulation of TREM2 activation could affect AD progression.

##### TREM2 Pathological Variants

TREM2 has an effect on increasing myeloid cell number in response to inflammation or disease; therefore, TREM2 deficiency is associated to a decrease of activated myeloid cells clustering in AD and to neurodegenerative diseases [106]. In the early stage of AD, microglia constitutes an envelope around the amyloid surface that delimits fibril development and contains amyloid plaques [107]. A lack of TREM2 or its ligand leads to dispersed amyloid plaques resulting in an increase in contacts with nerve structures [108].

Researchers [109] demonstrated reduced microglia cell accumulation around amyloid plaques in TREM-2 deficient AD mouse. Using murine models of 5xfAD, they found that if TREM2 is deficient, Aβ plaques appear more diffuse, less dense, and are associated with neuritic dystrophy. These results have demonstrated that TREM2 is required for the early expansion of microglia around Aβ plaques limiting their diffusion and the consequent amyloid-related neuronal damage. The failed clustering of microglia around amyloid plaques leads to excessive accumulation of Aβ at late stages of AD and build-up of dystrophic neurites around the plaques.

Therefore, even if microglial phagocytosis of Aβ may serve a neuroprotective function, the absence of TREM2 significantly impairs the ability of microglia to engulf amyloid plaques [108].

In contrast, overexpression of TREM2 facilitates Aβ1–42 phagocytosis and inhibits Aβ1–42-induced proinflammatory response [102]. Using a transgenic model of AD, Jiang et al. have demonstrated that TREM2 was upregulated in microglia under AD conditions and this upregulation was attributed to enhanced Aβ1–42 levels. They emphasize microglia modulation by TREM2 in AD and its dependence from DAP12. Overexpression of TREM2 reduces Aβ deposition, neuroinflammation, and neuronal loss with consequent amelioration spatial cognitive function [100]. In conclusion, the overexpression of TREM2 plays a protective role in both early- and mid-term AD, whereas this protective effect is lost in late-term AD [110]. In this framework, disruption of TREM2 activity was highlighted as an important risk of developing late-onset AD.

The role of TREM2 is widely studied, not only in AD, but different studies suggest that it may be involved also in the pathophysiology of multiple sclerosis (MS). A high level of soluble form of TREM-2 has been detected in the cerebrospinal fluid (CSF) of MS patients and has been proposed as a potential MS diagnostic biomarker [111]. Moreover, MS patients treated with natalizumab, an immune-modulating drug, showed an improvement in the clinical course correlated with a significant decrease in the CSF soluble TREM2 (sTREM2) supporting the crucial role of microglia in the pathophysiology of multiple sclerosis [112]. Indeed, the experimental autoimmune encephalomyelitis (EAE) mouse model of MS injected with anti-TREM2 monoclonal antibody (mAb) showed a reduced clearance of myelin and exacerbation in the disease severity [113]. Moreover, the TREM2 loss of function in mice treated with the curprizone (CPZ), a CNS demyelination model, induced increasing in the demyelinated lesions and reduced microglia phagocytic activity which was repristinated by TREM2 agonist antibody treatment which promoted the myelin debris removal by microglia and increased remyelination process [114].

Multiple human heterozygous rare variants in TREM2 have been linked to various neurodegenerative diseases and in particular R47H variant of TREM2 is one of the strongest single allele genetic risk factors that has been associated to AD [115]. It is characterized by a mutation of arginine in R47 to histidine by a DNA polymorphism [100]. Even if this variant has always been correlated to Nasu-Hakola disease (NHD) and to several cases frontotemporal dementia (FTD) [116,117], how it can increase this risk has been only recently described.

TREM2 R47H variant reduces affinity for TREM2 ligand binding and alters glycosylation leading to idea that the TREM2 R47H variant is a loss of TREM2 function [118]. This mutation decreases TREM2-dependent microglia activation and extracellular Aβ phagocytosis. Therefore, TREM2 insufficiency prevented proliferation and clustering of microglia around amyloid plaques facilitating neurodegenerative diseases. This led to accumulation of Aβ plaques at late stage of disease progression and increase of dystrophic neurites around the plaques [106]. TREM2 seems to have a protective function in 5XfAD mice [109].

In conclusion, given the involvement of TREM2 in the phagocytic role of microglia on amyloid plaques, the reduction of TREM2 activity in R47H variant may lead to brain damage through the inability of the brain to clear these toxic products [119].

##### Role of TREM2 in the Transition from Homeostatic to Pathological Microglia

R47H variant binds apolipoprotein E (ApoE) with less affinity than TREM2: ApoE modulation of microglial cells activation via TREM2 regulation participates in the neuronal loss in an acute model of neurodegeneration [120].

ApoE is the most abundant lipoprotein in the brain secreted primarily by astrocytes that plays an important role in lipid trafficking, cholesterol homeostasis, and synaptic stability [121]. In the brain, ApoE carries cholesterol and other lipids from astrocytes to neurons, where they are fundamental to maintain synaptic plasticity [122]. In addition, ApoE pathway modulates a switch from homeostatic to neurodegenerative microglia phenotype (MGnD) following phagocytosis of apoptotic neurons [120]. The disease-associated microglia downregulate homeostatic microglia genes and increase the expression of AD associated activation markers, such as APOE and Trem2 [123]. Different isoforms have been identified (apoE2, apoE3, apoE4) encode from the three common alleles (ε2, ε3, ε4). This differs in amino acid cysteine/arginine at position 158 leading to different binding affinity to ApoE receptors [121]. Among them, APOEε4 has largely correlated to AD progression confirming by a meta-analysis, in which APOEε4 carriers exhibit impaired episodic memory, executive function, and global cognition [124]. In this context, the ε4 allele of the APOE gene is the major known genetic risk factor [125] while the APOEε2 allele has a decreased risk, relative to the common APOE ε3 allele [126]. Moreover, APOE ε4 seems to exacerbate tau-mediated neurodegeneration, while the absence of this allele is protective [127].

APOEε4 is associated with synaptic degeneration and reduction in synaptic plasticity [128]. APOEε4 reduces neuronal surface expression of the LDLR family member ApoE Receptor 2, leading to suppression of synaptic transmission [129].

For a long time, APOEε4 and TREM2-R47H variants were identified as independent risk genes factors for late-onset AD (LOAD) [119]; however, recently, several studies have highlighted how the involvement of TREM2 in neurodegenerative disease through the modulation of microglia cells activation depends on ApoE [45]. TREM2 binding to ApoE lead to increase the phagocytosis of apoptotic neurons while TREM2 R47H variant reduces TREM2 affinity to bind ApoE [130]. When TREM2-ApoE pathway is activated, MGnD phenotype loses the ability to prevent neuronal loss and provide tolerogenic signals to T cells [120].

Despite the many questions remain open, the discovery that genetic polymorphisms of microglial immunoreceptors are crucial for developing AD allow to study inhibitory receptors as potential targets for therapeutic intervention.

## 4. Conclusions

Existing therapies for AD are for symptomatic AD and do not target the underlying etiopathogenic mechanisms. Currently, there is a strong demand for therapies able to interact with the pathogenic mechanisms involved in the neurodegenerative process and slow down disease progression. One such downstream target is neuroinflammation, which is known to represent a causal component rather than a consequence of neurodegeneration. These mechanisms are highly complex, and microglia represent the predominant modulators of neuroinflammation. Moreover, microglia play a critical role in inducing synapse loss and dysfunction, even though the molecular underpinnings remain elusive. Therefore, a better understanding of the cellular and molecular aspects underlying the microglia–synapse interaction is urgently needed for designing novel disease modifying approaches in AD. Whether preventing microglial-mediated elimination of synapses could prevent neurodegeneration and reduce cognitive decline warrants further investigations.

## Figures and Tables

**Figure 1 ijms-22-02342-f001:**
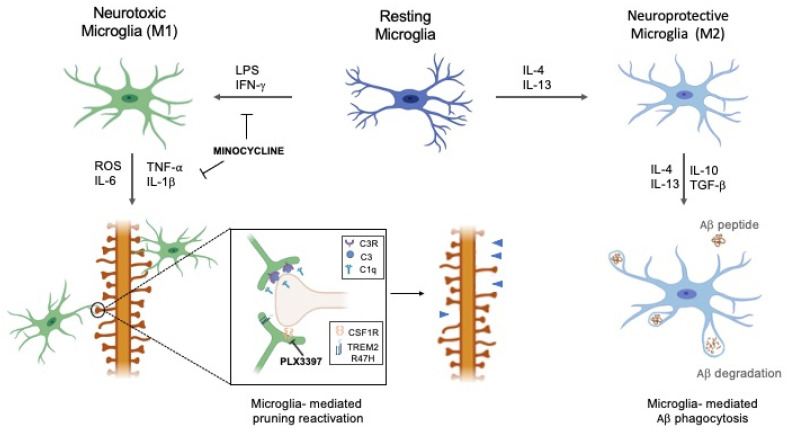
The dual role of microglial phenotypes in Alzheimer’s disease (AD). Depending on the received stimuli, resting microglia can shift to either neurotoxic (M1) or neuroprotective (M2) phenotype. Activated M2 microglia participate in the clustering of amyloid beta (Aβ) plaques and their consequent phagocytosis. Overproduction of pro-inflammatory cytokines by M1 can reactivate microglia-mediated pruning, aided by C1q and C3, leading to pathological synaptic loss. Minocycline can modulate the pathological activation of M1 phenotype and lock the pro-inflammatory mediators’ release, thereby reducing inflammation. The modulation of colony-stimulating factor 1 receptor (CSF1R) and triggering receptor expressed on myeloid cells 2 (TREM2) R47H receptors expressed by microglia, e.g., through CSF1R inhibitor Pexidartinib (PLX3397) administration, could exert a protective role in preventing microglia-mediated pruning reactivation.

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
