# Peer review of "Targeting Microglia-Synapse Interactions in Alzheimer’s Disease"

_ijms, 2021, doi:10.3390/ijms22052342_

Round 1
Reviewer 1 Report
In this current manuscript, Piccioni et al review the potential interest of targeting microglia-synapse interactions in Alzheimer’s disease (AD). Indeed, more and more data suggest the importance of the microglia in AD process though an uncontrolled acute neuroinflammation or synaptic pruning. In this context, deciphering molecular mechanisms involved has a great interest in the field.
Overall, this review is clearly written and correctly referenced. In particular, the first part of the review is a clear presentation of the hypothesis, supported by a nice figure, which is accessible for people who are not expert in the field.
My main concern is about the last part of this review (3.3. section) when the authors focus on CSF1R and TREM2 in order to review the potential mechanisms allowing modulation of microglial activation (senses 296-299). Of cause, I do not deny that CSF1R and TREM2 could play an important role, but there is not rational for focusing only on these two receptors. Microglial activation could be modulated by many other mechanisms.
For example, GWAS have identified many genes preferentially express in microglia which could control microglia activation like TREM2.
I suggest that additional paragraph should be added to discuss more widely the involvement of genetic risk factors or other pathway. Alternatively, the title of the review should be changed to clearly mention that authors will discuss the potential interest of CSF1R and TREM2 as therapeutic targets.
Reviewer 2 Report
The article "Targeting microglia-synapse interactions in Alzheimer's disease" submitted by Piccione et al., is well written, being deep enough to be a review.
Only minor revisions are necessary
- Authors must check that all abbreviations are properly named at the beginning of their first appearance.
- They should check the references and their numbering
- Some expressions in English are not very suitable.
